# Bridging the Knowledge Gap for Pressure Injury Management in Nursing Homes

**DOI:** 10.3390/ijerph19031400

**Published:** 2022-01-27

**Authors:** Ye-Na Lee, Dai-Young Kwon, Sung-Ok Chang

**Affiliations:** 1Department of Nursing, The University of Suwon, Hwaseong 18323, Korea; yenalee@suwon.ac.kr; 2Gifted Education Center, Korea University, Seoul 02841, Korea; dykwon.edu@gmail.com; 3BK21 FOUR R&E Center for Learning Health Systems, College of Nursing, Korea University, Seoul 02841, Korea

**Keywords:** nursing home, pressure injury, knowledge to action model, web-based education program

## Abstract

Background: Pressure injuries in nursing homes remain a consistent problem. Unfortunately, despite the variety of pressure injury education offered in nursing homes, the knowledge learned cannot be applied in practice, and as a result, the prevalence and incidence of such injuries are consistently high. This study aimed to address those gaps by analyzing the nursing competency for pressure injury management and implementing pressure injury education programs in nursing homes. Methods: Two phases were conducted based on the action cycle in the knowledge to action model. During the first phase, a framework was constructed by analyzing nursing experience. The second phase consisted of the implementation and monitoring of the program to evaluate the effects of the framework. Results: The main results for nursing competencies for pressure injury management in nursing homes are integrated thinking, understanding in an environmental context, interpersonal relationships for efficient decision making, and meeting any challenges to professional development. The results concerning the program’s effects showed significant differences in the participants’ knowledge, attitude, stage discrimination ability, and clinical management judgment ability. Conclusion: The educational framework and program derived from this study are expected to improve nurses’ pressure injury management competency in nursing homes and to contribute to effective pressure injury management and quality of life for residents in nursing homes.

## 1. Introduction

Due to the rapidly increasing number of aging societies worldwide, the population of older adults is expected to increase by 1 million every year [1]. It is also estimated that more than 100 million people over the age of 60 have chronic diseases, which has become a significant challenge for the health system [2]. As the demand for nursing care has increased, nursing homes that care for the elderly have also increased [3]. Therefore, interest in the quality of nursing care for the elderly in nursing homes is increasing, implying that the nurses working at these facilities need to improve their ability to care for them [4,5,6]. Specifically, the cause of pressure injuries in older adults is due to the lack of their self-nursing ability due to cognitive decline, loss of mobility, and loss of consciousness [3]. The elderly also present the most extrinsic factors for pressure injuries, such as limited mobility and reduced ability to perform daily activities [7].

Additionally, older adults are highly likely to be at risk of pressure injuries caused by internal factors due to decreased sensory function of the skin and functional damage to the skin owing to aging [8]. Notably, the prevalence of pressure injuries in nursing homes is higher than in hospitalized patients [7,9]. Since pressure injuries in older adults are an essential element in resident safety, the ability to manage for pressure injuries in the elderly is an integral part of improving the quality of nursing care [10,11,12]. An approach employing a multidisciplinary team consisting of professionals from various health and medical fields is vital for pressure injury management. Nonetheless, in preventing and evaluating pressure injuries, nurses play the most significant role, which is to assess residents’ risk factors and the changes in skin condition at an early stage [11,12,13]. This puts them in a crucial position in managing pressure injuries, which includes preventing, assessing, and providing scientific nursing care. In nursing homes, their competence is more important than in general hospitals because nurses are responsible for a significant proportion of direct nursing and decision-making tasks.

However, nursing homes have different requirements for general pressure injury management in terms of assessment and intervention due to the special needs of the residents, many of whom are experiencing cognitive decline, including dementia, or movement restrictions [3,8]. Additionally, facilities are limited in caring for pressure injuries due to a lack of human and material resources caused by problems such as frequent turnover [14,15]. Research has shown that nurses’ limited knowledge and experience with pressure injury management often arises from the lack of infrastructure at nursing homes and the characteristics of residents, indicating the need for proper comprehensive and practical education in this area [14]. A previous study revealed this problem was due to valid and reliable pressure injury risk assessment tools being seriously underused along with evidence-based management guidelines appearing to be rarely implemented in nursing homes [16]. Another study showed significant levels of underuse of knowledge, up to 28.4%, in pressure injury management, and these results suggest the necessity of an innovation education program to bridge the knowledge to action gap in medical practice [17]. Therefore, education for those working in nursing homes requires a strategy that focuses on imparting knowledge through practical education in the field.

Therefore, this study developed a customized education framework and pilot tested it to translate nurses’ knowledge of caring for pressure injuries at nursing homes into performance. Proper education on pressure injury management for institutions can lead to the improved practical performance of nurses at elderly nursing homes. Furthermore, it will improve the quality of pressure injury management practice and the overall quality of nursing care for the elderly living in such facilities.

## 2. Materials and Methods

### 2.1. Study Design

Developments in the current health system require continuous changes in the behavior of nurses, changes mainly addressed through continuing education [18]. Furthermore, there is a growing awareness that research findings are not making their way into practice. This has stimulated an increased interest in finding ways to minimize what is described as “the knowledge to action gap” [19]. The gap between knowledge and action presents a critical challenge within health system management and practice [20]. The knowledge to action (KTA) model is a thoroughly researched scientific model that was developed in Canada in the early 2000s after reviewing 31 planned action theories. In this study, the KTA model was used to develop interventions to reduce the gap between knowledge and performance to improve performance in nursing homes for pressure injury management (Table 1).

The first stage (phase 1) constituted preparing a framework for improving pressure injury management performance through identifying problems, adapting knowledge to the local context, and assessing barriers to using the knowledge from the KTA model performance stages. In the second stage (phase 2), the effectiveness of the educational program developed based on the framework was verified by selecting, tailoring, and implementing the interventions and by monitoring knowledge use, evaluating outcomes, and sustaining knowledge use during the program’s execution.

### 2.2. Phase 1: Framework Composition to Enhance the Performance of Caring for Pressure Injuries in Nursing Homes

#### 2.2.1. Problem Identification

In-depth interviews were conducted with 10 nursing practitioners in nursing homes to identify the homes’ problems in terms of caring for pressure injuries. Participants with practical experience who could provide sufficient data on the subject matter were selected through purposeful sampling, and repeated interviews were conducted until theoretical saturation was reached. The inclusion criteria were nurses with at least one year of experience working in a nursing home, and the exclusion criteria were nurses who were currently inactive. Interviews were conducted for approximately 60 minutes per session, and an average of two interviews were conducted per participant. Interviews were recorded with the consent of the research participants, and the researcher made field notes to record the main situations and meaningful content. The data analysis was performed through content analysis [21]. The in-depth interview materials were read repeatedly, and similar topics were organized and categorized to identify problems.

#### 2.2.2. Adapting Knowledge to the Local Context

A Consensus-Oriented Decision-Making (CODM) discussion was conducted to understand the educational design suitable for the management of pressure injuries in nursing homes [20]. For the CODM discussion, participants suitable for the study were intentionally sampled, targeting professionals at the nursing home. The inclusion criterion was being an expert in practice with more than three years of pressure injury management experience in a nursing home. The group formed comprised 6 people (the preferable limit is less than 10). The participants were very knowledgeable about the topic, and a small group was more efficient than a large group since the topic was discussed on a professional basis [22]. The exclusion criterion was those who did not work at the facility.

CODM is a decision-making model that leads to group consensus, and interviews were conducted according to the model’s steps. From selecting the topic to the decision-making, the researcher provided guidance and facilitated an open discussion. Furthermore, the researcher selected the primary topic while clarifying the explicit criteria for satisfactorily solving the participants’ concerns. Once a proposal for each primary topic was formed, a specific draft was integrated for the participants’ consensus in reaching a formal decision. To analyze the data derived through CODM, Ritchie and Spencer’s framework analysis method, which is a qualitative analysis method, was used [23].

#### 2.2.3. Assessing Barriers to Knowledge Use

The disability and facilitation factors were confirmed through content validity verification by five nursing home experts and five pressure injury management experts [24]. Participants were selected through purposeful sampling of the subject matter, and the inclusion criterion was being an expert in practice with more than five years of nursing home or wound management experience. Based on this, the final framework was completed.

### 2.3. Phase 2: Development of Educational Program Based on the Configured Framework and Verification of Effectiveness

#### 2.3.1. Selection, Tailoring, and Implementation of Interventions

An educational program was designed based on the developed framework, and its effects were confirmed (Table 2). The educational contents were composed of evidence-based guidelines to enhance integrated thinking ability. Simultaneously, cases with the elderly, bedridden residents, and dementia residents were used in refining the teaching method to enhance the program’s environmental context. Furthermore, education based on standardization tools related to pressure injuries was provided to accentuate the nurse’s role as an efficient intermediary. In parallel to this, academic journal searches and conferences were introduced to construct an improved, knowledge-based human infrastructure. Finally, a computer educator and a nursing professor implemented a web-based system for continuous use (Figure 1).

#### 2.3.2. Monitoring Knowledge Use, Evaluating Outcomes, and Sustaining Knowledge Use


(1)Participants


The subjects of this study were 35 nurses working in nursing homes. The inclusion criteria for selecting research subjects were: (1) those who had worked at a nursing home for at least one year and (2) those who had consented to participate in the study. The exclusion criterion was nurses who are currently not practicing. The sample size was determined to be 30 subjects for an effect size of 1.06 based on the suggestions from Cohen and a previous study that educated nurses on pressure injuries [25] using the G*Power 3.1.9.2 program, calculated for the power 0.8 and significance level 0.05. However, considering a 30% dropout rate, 40 subjects were recruited to be divided into 2 groups, an experimental and a control, of 20. For the final analysis, the data from 17 people in the experimental group and 18 in the control group were used, excluding the dropouts who did not complete the training and questionnaire within the given period. The participants were all female, and the age ranges of the experimental group were 25–40 years old (n = 4, 23.53%), 41–50 years old (n = 5, 29.41%), and 51–61 years old (n = 8, 47.06%). The control group included nurses who were 25–40 years old (n = 2, 11.11%), 41–50 years old (n = 5, 27.78%), and 51–61 years old (n = 11, 61.11%). The average total nursing experience was 13.94 ± 7.30 years for the experimental group and 15.89 ± 8.63 years for the control group. Finally, the experimental group’s average nursing experience in nursing homes was 4.53 ± 4.57 years and the control group’s was 3.72 ± 2.27 years.
(2)Procedure

After confirming the participants’ eligibility, the research objectives and methods were explained, and they were given the research participation consent form. It was explained that the subject’s anonymity was guaranteed and that personal information would not be used for purposes other than the present research. Participants were informed that consent could be withdrawn at any time while participating in the study. Once the consent form was completed, data were collected by the researcher using a structured questionnaire. The program developed in this study was applied on the experimental group while the control group received general education for pressure injury management. The education for both groups was conducted for 120 min. The pre-survey questionnaire was distributed after education and used as the post-survey. A follow-up survey was conducted for the experimental and control groups using the same questionnaire four weeks after the end of the intervention program.
(3)Outcome measures

Pressure injury management knowledge

To measure the knowledge of pressure injury management, the Pieper–Zulkowski Pressure Ulcer Knowledge Test (PZ-PUKT) developed by Pieper and Zulkowski (2014) [26], and supplemented and revised by Park (2018), was used with permission [25]. It consists of 39 questions: 19 questions about the pressure injury stage check, 9 questions concerning wound assessment, and 11 questions on dressing methods. Each question can be answered as “yes,” “no,” or “do not know” and is given 1 point for correct answers and 0 points for incorrect answers. The total score ranges from 0 to 39, with higher scores indicating better knowledge of pressure injuries. At the time of this tool’s development, its internal consistency reliability (Cronbach’s α) was 0.80 in Piper and Zulknowski’s (2014) study and 0.70 in Park’s (2018) study [25,26]. The Cronbach’s α for the present study was 0.81.

Pressure injury management attitude

A tool developed by Beeckman et al. (2010) [27] and modified and supplemented by Lee et al. (2014) [28] to measure pressure injury management attitude was used with permission after further modification and supplementation by Park (2018) [25]. This tool consists of 11 questions in 5 areas: 2 questions on the responsibility for pressure injury management, 2 questions on priority selection, 2 questions on the effects of a pressure injury on patients and the environment, and 3 questions on self-confidence. Each item is scored from 1 point (“not at all”) to 4 points (“strongly agree”), and the overall score ranges from a minimum of 11 points to a maximum of 44 points, with higher scores indicating a more positive attitude toward pressure injury management. The internal consistency reliability of this tool in Park’s (2018) study was 0.70, and its internal consistency reliability in this study was 0.76.

Ability to identify pressure injury stages

A tool developed by Lee et al. (2016) was used with permission for pressure injury stage identification ability [29]. The tool contains 23 photos and clinical information of patients, such as medical diagnosis, patient mobility, defecation status, cervical fluid injection, and wound retention period. The 23 photos comprise 18 photos of each pressure injury stage, including medical device-related pressure injuries, from 6 six stages of pressure injury classification, 2 photos of pale erythema, and 3 photos of incontinence-related dermatitis. A correct answer is given 1 point, while an incorrect answer is converted to 0. The total score ranges from 0 to 23 points, where a higher score indicates a higher degree of visual identification ability. The reliability of the tool at the time of development was K-R 20 = 0.81. The internal consistency reliability of the tool in this study was 0.72.

Clinical judgment on pressure injury management

The ability to perform clinical judgment on pressure injuries was defined by a score measured using Lee’s (2014) tool [28], which was a modification of the clinical judgment tool developed by Lasater (2007) [30] for use related to pressure injuries. Classification items for each domain were configured to be converted to a scale of 1 to 4 points, with the lowest total score being 11 points and the highest 44 points. Higher scores indicated a remarkable ability to make a clinical judgment. A score of 11 is defined as the beginner stage; a score of 12 to 22 is defined as the development stage; 23 to 33 is referred to as the achievement stage; and 34 to 44 is considered the proficiency stage. In the study by Lee (2014), Cronbach’s α was 0.75, and the Cronbach’s α for this study was 0.79.
(4)Data analysis

The data collected in this study were analyzed using the SPSS Statistics Windows program version 25.0. The statistical significance level selected was *p* = 0.05. Measurement tool reliability was tested with the Cronbach alpha coefficient. Descriptive statistics were used to determine the general characteristics of subjects, pressure injury knowledge, pressure injury attitude, pressure injury identification ability, and the ability for the clinical judgment of pressure injury management. An ANOVA, chi-squared test, Fisher’s exact test, Mann–Whitney U test, and independent *t*-test were used for general characteristics and pressure injury management-related variables to test for homogeneity between the two groups. The effect of the education program was analyzed using a repeated measure ANOVA and *t*-test. If the sphericity assumption was not satisfied, the modified Greenhouse-Geisser correction value was checked.

Consequently, all were found to measure 0.90, confirming that they could be substituted for sphericity. For post hoc testing, a Bonferroni analysis was used, and if even one variable was not normally distributed, it was analyzed by a Friedman test. Finally, the keystone variables between the time points of the experimental group and the control group were analyzed using a repeated measure ANOVA.

## 3. Results

### 3.1. Phase 1: Composition of the Framework to Enhance the Performance of Pressure Injury Management in Nursing Homes

From the in-depth interviews, unpredictable wound healing process, restrictive situation(s), confusion in the access system, and unstable support system were derived as the obstacles for pressure injury management in care facilities. In solving these problems in CODM, which was carried out based on this, it was agreed that integrated thinking, understanding in environmental contexts, an intermediary role for efficient decision making, and securing a perspective for improvement were necessary. As a result of the expert validity verification of the agreed-upon items, it was confirmed that this framework was properly composed by measurements of 4.94 ± 0.25, 5.00 ± 0.00, 4.94 ± 0.25, and 5.00 ± 0.00 for each domain (Figure 2).

#### 3.1.1. Unexpected Wound Healing Process: Integrated Thinking

Participants noted that they felt confused while observing the pressure injury healing process, which differed from their education knowledge. Therefore, it was emphasized that they should develop integrated thinking to lead pressure injury management. Such thinking should include clearly understanding individual circumstances and re-judging assessments in a universal framework and then adjusting to the circumstances, as individual facility resident characteristics are unique and diverse. Representative participants’ statements are included below:


*“I tried the products I received as a sample from the training before, but they did not work. Thus, when I asked, they said that it was difficult to see good effects on residents who are in poor condition.”*



*“Despite knowing that prevention is paramount for pressure injuries, there are too many other major problems for older people. Pressure injuries are often not even listed among the priorities because the elderly have many life-threatening problems, such as when a resident has a breathing problem and has to remain in a position that causes pressure injuries, knowing that it would cause pressure injuries.”*



*“There are times when things go wrong, particularly when I am obsessed with a product that was recommended for use by the hospital, thinking that it is the only answer. However, if something does not feel right, it is better to reassess from the beginning to make a judgment or ask for help. I think it is important to re-think the whole thing according to that specific pressure injury and the resident’s condition on that day.”*


#### 3.1.2. Restrictive Situation: Understanding in an Environmental Context

It was mentioned that there are difficulties because most residents in nursing homes are older adults with weak skin integrity. Furthermore, they are at risk of developing and aggravating pressure injuries due to physical and mental problems, such as cognitive impairment and supine bed position. It was also noted that the items or treatment tools that can be used are limited because the facilities do not have specialized medical departments. Therefore, it was emphasized that facility nurses should understand the characteristics of residents and make assessments and interventions accordingly. It was also mentioned that it is necessary to have an overall understanding of products or tools and effectively apply them by identifying the pros and cons of alternative products. Representative participants’ statements are included below:


*“I believe most of the nurses in the facility have some knowledge. However, unusual cases also do occur. Occasionally, unusual cases occur depending on the resident’s degree of contraction and mobility.”*



*“I received education on pressure injury management, but not all the products I learned about at that time are available at the facility.”*



*“Not only do residents have diverse economic conditions but if the resident’s condition, for example, is near the end of the life cycle, there are cases in which a low-cost product is preferred over a high-priced product. In that case, you need to know about the various items that can be used as an alternative.”*



*“Residents who have dementia may have problem behaviors, and the medications they take may cause them to spend more time in bed. In that’s the case, although they are not in the high-risk group for pressure injuries, pressure injuries may develop.”*


#### 3.1.3. Confusion in the Access System: Interpersonal Relationships for Efficient Decision-Making

Participants mentioned that having multiple caregivers who make decisions about residents, some of whom are often moving in and out of the facility to receive advanced care at a hospital or clinic, were possible obstacles. Additionally, this presents unsystematic access for practitioners from various fields concerning collaboration. Therefore, the need for resident-centered, multidisciplinary cooperation was emphasized for activating the management of pressure injuries in facilities. To achieve this, it is necessary for multidisciplinary caregivers to understand each other’s roles and to establish a cooperative system that promotes sharing and interaction. Furthermore, older adults or residents with cognitive impairment often must make decisions with the help of their caregivers. Thus, practical communication skills were mentioned as essential to setting goals for wound management together with the caregivers and to increase their participation. Representative participants’ statements are included below:


*“Even if we explain the resident’s pressure injuries well to the caregiver, there are cases where continuous care cannot be provided when a different caregiver comes later. If communication with the caregiver is not effective, pressure injuries cannot be treated properly.”*



*“I asked for pressure to not be applied on pressure injuries while the resident is in physical therapy, but this was not well communicated, and there were times when pressure was applied.”*



*“Sometimes residents need to go to the hospital for pressure injury treatment. In these cases, written information is received, but the caregiver is responsible for relaying the information from the hospital accurately; treatment is difficult when this is not done.”*



*“Nurses, social workers, physical therapists, and nutritionists all get together and have a meeting about pressure injuries once a week. Based on the results, if each of us put in the effort, such as performing physical therapy while avoiding the area with pressure injuries as much as possible, or preparing nutritious food for healing, good results were obtained.”*


#### 3.1.4. Unstable Support System: Meeting any Challenges to Professional Development

The participants conveyed that they experienced difficulties in treating pressure injuries due to changes such as frequent personnel turnover and the absence of professional medical staff for wound healing compared to hospitals. In this unstable situation, they expressed that it is necessary to increase the ability to develop individual competency in pressure injury management, such as searching for data or participating in education. If it is impossible to solve the problem with individual competence, they mentioned that management could be effective, efficient, and sustainable only when the network could be systematically constructed. For that, they should apply the information obtained from external sources flexibly, according to the facility. Representative participants’ statements are included below:


*“Each nurse at our facility has a highly varying degree of competence. However, even the assessment of pressure injuries can sometimes be incorrect when assessing the same pressure injury, and it is challenging for us to perform interventions accordingly.”*



*“As we do not have a doctor specializing in wound healing, nurses often have to make decisions themselves, leaving us feeling anxious.”*



*“As there are a variety of professions working together at the facility, it is necessary to consider pressure injury management as something important and have the ability to motivate everyone involved. Not only knowledge but responsibility, attitude, and attention are the keys to success.”*



*“I met with a wound specialist at a nearby teaching hospital and kept in contact with them and asked for advice. Nonetheless, because the environment is different, we cannot apply it as-is. Still, we are working hard to establish a connection by applying the best possible method in our facility and sending residents to a hospital for treatment if necessary.”*


### 3.2. Phase 2: Development of the Educational Program Based on the Configured Frame and Verification of Effectiveness

#### 3.2.1. Prior Homogeneity Verification of Subjects’ General Characteristics and Dependent Variables

The homogeneity verification of the experimental and control groups’ general characteristics found no significant differences in gender, age, academic background, nurses’ overall working experience, and work experience in a nursing home. Furthermore, homogeneity between the two groups was confirmed. No significant differences in the characteristics of pressure injury management, such as experience, comfort and confidence, and education, were found. Additionally, as a result of verifying the homogeneity of the dependent variables of the experimental group and the control group, there was no statistically significant difference in pressure injury management knowledge (t = 0.27, *p* = 0.79), pressure injury management attitude (t = −0.55, *p* = 0.59), and pressure injury stage identification ability (t = −0.17, *p* = 0.87). The ability for clinical judgment on pressure injury management (t = −0.18, *p* = 0.86) also confirmed the homogeneity between the two groups.

#### 3.2.2. Verification of the Effectiveness of the Education Program

The pressure injury management knowledge scores were 26.12 ± 3.35, 29.76 ± 4.58, and 31.41 ± 3.79 for the experimental group and 25.72 ± 5.00, 26.50 ± 4.59, and 27.89 ± 2.45 for the control group before the intervention, immediately after the intervention, and four weeks after the intervention, respectively (Table 3). As a result of analyzing this by a repeated measures ANOVA, a significant difference in the change in the knowledge level among the three time points was found. This was the main effect (F = 15.894, *p* < 0.001). There was also a significant difference in the change in the knowledge level between the groups (F = 4.431, *p* = 0.043), along with a significant difference in the interaction effect between the measurement period and the groups (F = 3.402, *p* = 0.039).

For the experimental group, the pressure injury management attitude score was 35.00 ± 4.85 before the intervention, 36.29 ± 5.63 immediately after the intervention, and 40.47 ± 2.94 four weeks after the intervention; in the case of the control group, the scores were 35.72 ± 2.54, 36.06 ± 4.12, and 36.94 ± 4.29, respectively (Table 3). As a result of analyzing this with a repeated measures ANOVA, there was a significant difference found in the level of attitude among the three time points. This was the main effect (F = 10.193, *p* < 0.001), while there was no significant difference in the change in the degree of attitude between the groups (F = 0.853, *p* = 0.362). However, there was a significant difference in the interaction effect between the measurement period and the groups (F = 4.152, *p* = 0.020).

In regard to the ability to identify pressure injury stages, the scores were 9.94 ± 3.94, 13.76 ± 2.99, and 15.47 ± 2.81 for the experimental group and 10.17 ± 4.12, 10.83 ± 3.09, and 11.83 ± 2.98 for the control group before the intervention, immediately after the intervention, and four weeks after the intervention, respectively (Table 3). As a result of a repeated measures ANOVA analysis, a significant difference was found in the degree of pressure injury stage identification ability among the three time points. This was the main effect (F = 30.647, *p* < 0.001), and the degree of pressure injury stage identification ability between groups also showed a significant difference (F = 4.443, *p* = 0.043). Furthermore, there was a significant difference in the interaction effect between the measurement period and the groups (F = 9.814, *p* < 0.001).

Concerning the clinical judgment on pressure injury management, the scores were 23.82 ± 2.70, 27.18 ± 2.30, and 30.65 ± 3.00 for the experimental group and 24.00 ± 3.12, 25.56 ± 2.59, and 25.94 ± 3.62 for the control group before the intervention, immediately after the intervention, and four weeks after the intervention, respectively (Table 3). Analyzing this with a repeated measures ANOVA showed a significant difference in the change in the clinical judgment on pressure injury management among the three time points. This was the main effect (F = 32.197, *p* < 0.001). There was also a significant difference in the clinical judgment on pressure injury management between the groups (F = 7.295, *p* = 0.011), as well as a significant difference in the measurement time and the interaction effect between the groups (F = 10.152, *p* < 0.001).

## 4. Discussion

This study aimed to develop a framework that can improve pressure injury management competency in nurses at nursing homes so that they can perform proper nursing on pressure injuries at the facility. The study also verified the effectiveness of a program implementing this framework. Since consistency in education is essential for the standardization of nursing work, it is necessary to present standardized practical education as a countermeasure for accurate and effective pressure injury management performance [10]. However, the current pressure injury management education is mainly designed to implement prophylaxes and undertake therapeutic activities related to treatment in general hospitals. Thus, it is difficult for nurses in the different environments of nursing homes to apply the training directly in practice. Therefore, in this study, a pressure injury management practice education program that considered the environmental dimension of nursing homes and that focused on improving the nursing competency of nurses working in such facilities was constructed and developed.

Existing education does not reflect the characteristics of vulnerable nursing home residents, such as dementia and supine bed position, and the characteristics of facilities with limited resources, making it challenging for its application in practice. When the previous studies on pressure injury education in nursing homes were examined, many of them focused on prevention only, and the characteristics of nursing homes that require management of unavoidable pressure injuries in the vulnerable elderly were not reflected [10,31]. Furthermore, these long-duration programs could not be applied to nursing homes with a lack of nursing personnel [32]. Unlike existing educational programs, the present one is structured based on strategies and contents derived from in-depth interviews with nursing practitioners and CODM for pressure injuries in nursing homes. It focuses on prevention and is comprehensively applicable in such facilities. The gap between theoretical content and practice was filled with an educational program for understanding the field and compiling experts’ opinions in practice.

The experimental group who received the pressure injury management practice training developed in this study showed a significant difference from the control group who received the pressure injury management training for general nurses. In particular, this was seen in the interaction effect between measurement time points and groups in all hypotheses for pressure injury management knowledge, pressure injury management attitude, pressure injury stage identification ability, and the ability for clinical judgment for pressure injury management. These findings suggest that the practical education program developed according to the KTA model in this study could be used in practice at facilities, as it was composed according to the context of nursing homes and that the knowledge obtained from training could be developed further and retained with use [33]. Furthermore, since the knowledge was relatively consistently retained, even after four weeks from intervention, this education is significant compared to that in studies that showed decreasing knowledge over time after the completion of training [34]. Notably, compared to existing studies that mention a low level of knowledge in pressure injury management at nursing homes, the results are significant because the experimental group’s knowledge level became comparable to that of hospital nurses [35,36]. Enhanced knowledge is even more significant since inadequate knowledge in pressure injury management at nursing homes has been considered an obstacle in previous studies [14].

Although there was no significant difference in attitude before and after this training, the significant increase after four weeks could be regarded as the impact of this educational program in increasing the perception of the importance of pressure injury management among nurses and improving their attitude for selecting and performing interventions for the treatment of pressure injuries. This could present a solution to the problem cited by previous studies that mentioned a lack of confidence in pressure injury management due to the lack of education among the nurses working in nursing homes [14]. Furthermore, based on previous studies, which reported that nursing ability is enhanced when nurses have confidence instilled through a positive perception of nursing, the changes toward positive attitude on pressure injuries through this study could be regarded as a significant result [37].

Identifying the pressure injury stage and making a clinical judgment is critical in directly performing pressure injury management in practice and is often the area in which nurses have difficulty. The sound clinical judgment of nurses plays the intermediary role of transferring knowledge to performance and is an essential variable in converting knowledge learned from pressure injury education to practice [12]. Therefore, the improvement made through this study would be highly significant in terms of the effect of the developed program.

In particular, while a previous study that evaluated the knowledge, attitude, and performance of nurses working in a long-term care facility in Korea found that their levels of knowledge and positive attitude were average, it has been shown that their performance was lacking due to, for example, consulting the pressure injury management nursing plan to review the plan rather than focusing on the patient’s condition. The results of this study are expected to serve as a strategy to resolve such problems [35].

Although this study was conducted as a pilot study with a few subjects to develop a framework and examine its effect, the framework was verified with significant results [29,33]. The framework and educational program developed through this study were designed to enhance the competence of nurses in pressure injury management at nursing homes. Nevertheless, we suggest that the program be developed further into a continuous or expert program suitable for improved pressure injury management ability. For continuous application, we expect that future studies could show the results in connection to the quality of life in residents following the treatment of pressure injuries to create a periodic national training with government support or mandatory training for the management of elderly care facility quality.

## 5. Conclusions

This study was conducted to develop and provide a framework for improving pressure injury management competency in nurses working in nursing homes. The results of this study support the KTA model finding that the quality of care at such facilities depends on nurses’ pressure injury management competence. Nurses at such facilities need to have integrated thinking ability, an understanding of the environmental context, communication skills for efficient decision making, and a vision for systematic improvement for patient pressure injuries to be successfully managed. This framework should serve as the primary data for professional guidelines for experienced and novice nurses at long-term care facilities. To enhance nurses’ competence, the educational program utilizing this framework should be extended and established as standardized pressure injury training. The results of this study are expected to improve the safety and quality of life of the elderly in nursing homes.

## Figures and Tables

**Figure 1 ijerph-19-01400-f001:**
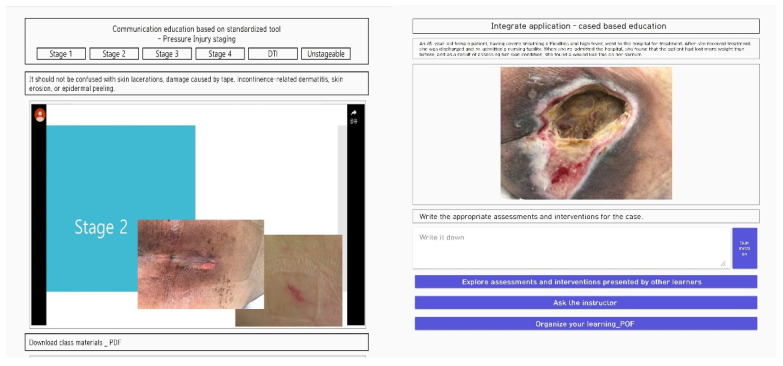
Example illustrations of an educational program for pressure injury management in nursing homes.

**Figure 2 ijerph-19-01400-f002:**
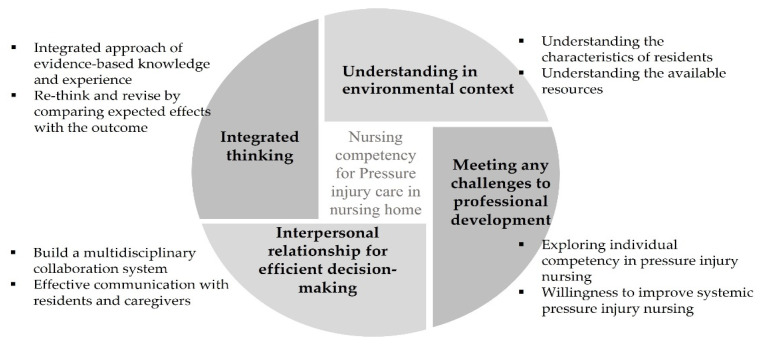
The framework of nursing competency for pressure injury management in nursing homes.

**Table 1 ijerph-19-01400-t001:** Components of the KTA model, with methods used.

Components of KTA Model	Methods Used to Address Component in This Study		
	Subject	Study Method	Objective
Problems identified	A total of 10 nurses at a nursing home	In-depth interview	Identification of composition and problems of pressure injury management knowledge at nursing homes
Adapted to the local context	Six current working nurses at a nursing home	CODM	To design a practical education framework suitable for pressure injury management in nursing homes, we agreed on a strategy and education composition for pressure injury management in nursing homes.
Barrier assessment	Five current workers at a nursing home and five wound nurses	Content Validity Index (CVI)	Barriers and facilitating factors were identified through content validity verification.
Selection, tailoring, and implementation of interventions	Two researchers and one computer program developer	Program development	We developed a web-based pressure injury management educational program for nurses in nursing homes.
Monitoring evaluation	A total of 35 nurses at nursing homes (17 nurses in the experimental group, 18 nurses in the control group)	Study design with randomizedexperimental and control group	Tests were conducted for nurses in nursing homes on their knowledge, attitude, pressure injury stage identification ability, and the ability for clinical judgment on pressure injury management.
Sustaining knowledge use			After 4 weeks of intervention, the same test was conducted to confirm continuity.

**Table 2 ijerph-19-01400-t002:** Interventions for barriers to integrating an educational program.

Problems Identified		Adapting Knowledge to the Local Context/ Assessing Barriers to Knowledge Use		Implementation in an Education Program
Unpredictable wound healing process	Limitations of generic pressure injury education based on the normal healing process	Integrated thinking	Integrated approach of evidence-based knowledge and experience	Enhance educational content based on the guidelines
	Problems of diseases with higher priorities than pressure injuries		Re-think and revise by comparing expected effects with the outcome	
Restrictive situations	Characteristics of residents vulnerable to pressure injuries	Understanding in environmental contexts	Understanding the characteristics of residents	Case-based education
	Limited resources at nursing homes		Understanding the available resources	
Confusion in the access system	Variability in the person deciding for the resident	Interpersonal relationships for efficient decision making	Build a multidisciplinary collaboration system	Communication education based on standardized tools
	Lack of organized system		Effective communication with residents and caregivers	
Unstable support system	Limited personnel and frequent turnover	Meeting any challenges to professional development	Exploring individual competency in pressure injury management	Introduction accessible network
	Lack of personnel to provide professional knowledge		Willingness to improve systemic pressure injury management	

**Table 3 ijerph-19-01400-t003:** Effectiveness of the educational program.

Variables	Time	Exp.	Cont	Source	F (*p*)
		MD ± SD	MD ± SD		
Pressure injury nursing knowledge	Pre-test	26.12 ± 3.35	25.72 ± 5.00	Group	4.43 (0.04)
	Post-test	29.76 ± 4.58	26.50 ± 4.59	Time	15.89 (<0.001)
	Follow up	31.41 ± 3.79	27.89 ± 2.45	GxT	3.40 (0.04)
Pressure injury Nursing attitude	Pre-test	35.00 ± 4.85	35.72 ± 2.54	Group	0.85 (0.36)
	Post-test	36.39 ± 5.63	36.06 ± 4.12	Time	10.19 (<0.001)
	Follow up	40.47 ± 2.94	36.94 ± 4.29	GxT	4.15 (0.02)
Ability to identify pressure injury stages	Pre-test	9.94 ± 3.94	10.17 ± 4.12	Group	4.43 (0.04)
	Post-test	13.76 ± 2.99	10.83 ± 3.09	Time	30.65 (<0.001)
	Follow up	15.47 ± 2.81	11.83 ± 2.98	GxT	9.81 (<0.001)
Clinical judgment on pressure injury management	Pre-test	23.82 ± 2.70	24.00 ± 3.12	Group	7.30 (0.01)
	Post-test	27.18 ± 2.30	25.56 ± 2.59	Time	32.20 (<0.001)
	Follow up	30.65 ± 3.00	25.94 ± 3.62	GxT	10.15 (<0.001)

## Data Availability

Data that support the findings of the study are available upon reasonable request from the corresponding author.

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
