# Peer review of "Bridging the Knowledge Gap for Pressure Injury Management in Nursing Homes"

_ijerph, 2022, doi:10.3390/ijerph19031400_

Round 1
Reviewer 1 Report
The introduction and discussion section should be reinforced. It is important to improve these sections with references and current studies that work on the subject.Author Response
[p2:Introduction; p13:Discussion]
The introduction and discussion section should be reinforced. It is important to improve these sections with references and current studies that work on the subject.
Response:
Thank you very much to the commnet. Regarding the issue, we reinforced the introduction section and the discussion section with refereces and current studies.
Introduction :
A previous study revealed this problem was due to valid and reliable pressure injury risk assessment tools being seriously underused along with evidence-based management guidelines appearing to be rarely implemented in nursing homes [16]. Another study showed significant levels of underuse of knowledge, up to 28.4%, in pressure injury management, and these results suggest the necessity of an innovation education program to bridge the knowledge to action gap in medical practice [17].
Discussion:
In particular, while a previous study that evaluated the knowledge, attitude, and performance of nurses working in a long-term care facility in Korea found that their levels of knowledge and positive attitude were average, it has been shown that their performance was lacking due to, for example, consulting the pressure injury management nursing plan to review the plan rather than focusing on the patient's condition. The results of this study are expected to serve as a strategy to resolve such problems [35].
Reviewer 2 Report
Introduction: It is recommended to change pressure ulcers to pressure injuries, not all lesions are ulcers, according to the latest international recommendations of the main scientific societies.
Methods: Were the same 10 nurses who participated in phase 1 the same nurses who participated in the Assessing barriers to knowledge use phase? Please clarify.
Author Response
[Overall]
Introduction: It is recommended to change pressure ulcers to pressure injuries, not all lesions are ulcers, according to the latest international recommendations of the main scientific societies.
Response:
Thank you very much for the comment. Regarding the issue, we changed pressure ulcers to pressure injuries
[p.4: Methods-2.2.3]
Methods: Were the same 10 nurses who participated in phase 1 the same nurses who participated in the Assessing barriers to knowledge use phase? Please clarify.
Response:
Thank you very much for the comment. Regarding the issue, we added the inclusion criteria for participants.
=> Participants were selected through purposeful sampling on the subject matter, and the inclusion criterion was being an expert in practice with more than five years of a nursing home or wound management.
Reviewer 3 Report
This is an interesting work on the impression of prevention and wound healing systems. Nevertheless, it needs to be improved and slightly rebuilt in my opinion. I am a clinician and I am interested in nursing in the world. I do not know what competences a nurse in Korea has in the field of wound healing, whether she should have a course, specialization or just professional experience. In the methodology, I did not find the characteristics of the groups, besides, the study protocol is described in a difficult language and is not clearly legible - it requires a lot of concentration from the reader, it would be worth rethinking and simplifying it a bit. is the correct formula, we try not to use such a language in the country where she practices, The nurse undertakes curative and therapeutic activities in the care and care of the patient. Conclusions should be structured differently

Author Response
[p.1; Abstract]
'Unfortunately, gaps exist in the knowledge of how to manage" - the formula is too general, it should be clarified
Response:
Thank you very much for the comment. Regarding the issue, we changed and added the details.
=>Unfortunately, despite the variety of pressure injury education offered in nursing homes, the knowledge learned cannot be applied in practice and, as a result, the prevalence and incidence of such injuries are consistently high.
[Overall]
According to the global NPIAP / EPUAP (Kottner et. al. 2019) guidelines, define pressure injury
Response:
Thank you very much for the comment. Regarding the issue, we changed pressure ulcers to pressure injuries
[p.1: Abstract and Overall]
" integrated thinking ability, the ability to share understanding in an environmen"- are these nursing competences ? maybe it should be called differently, I have doubts as to the nomenclature
Response:
Thank you very much for the comment. Regarding the issue, we changed the naming of nursing competency for pressure injury managmenet in nursing home.
=> The main results for nursing competencies for pressure injury management in nursing homes are integrated thinking, understanding in an environmental context, interpersonal relationships for efficient decision-making, and meeting any challenges to professional development.
[p.13; Discussion]
as a reader and clinician, I would like to know what the Korean nurses have in the prevention and treatment of pressure ulcers, it should be presented in the form of a table or specifically described in the discussion section
Response:
Thank you very much for the comment. Regarding the issue, we added the study that evaluates the knowledge, attitude, and performance for pressure injury management in korea.
=> In particular, while a previous study that evaluated the knowledge, attitude, and performance of nurses working in a long-term care facility in Korea found that their levels of knowledge and positive attitude were average, it has been shown that their performance was lacking due to, for example, consulting the pressure injury management nursing plan to review the plan rather than focusing on the patient's condition. The results of this study are expected to serve as a strategy to resolve such problems [35].
[p.2: Materials and Methods-2.1.Study design]
Please describe from general to detail, this description is not very understandable for the reader
Response:
Thank you very much for the comment. Regarding the issue, we changed and added the sentence according to your comments.
=> Developments in the current health system require continuous changes in the behavior of nurses, changes mainly addressed through continuing education [18]. Furthermore, there is a growing awareness that research findings are not making their way into practice. This has stimulated an increased interest in finding ways to minimize what is described as “the knowledge to action gap” [19].
[p.6: Materials and Methods-participants-2.3.2 (1)]
I do not see the description of the sample characteristics of the experimental group (basic and control)
Response:
Thank you very much for the comment. Regarding the issue, we added the characteristics of participants.
=> The participants were all female, and the age ranges of the experimental group were 25-40 years old (n=4, 23.53%), 41-50 years old (n=5, 29.41%), and 51-61 years old (n=8, 47.06%). The control group included nurses who were 25-40 years old (n=2, 11.11%), 41-50 years old (n=5, 27.78%), and 51-61 years old (n=11, 61.11%). The average total nursing experience was 13.94±7.30 years for the experimental group and 15.89±8.63 years for the control group. Finally, the experimental group’s average nursing experience in nursing homes was 4.53±4.57 years and the control group’s was 3.72±2.27 years.
[Overall]
Does Korea use the term bedsores care or pressure injury treatment?
correct nomenclature should take into account treatment or management, care refers to the patient not to the pathology of the wound
Response:
Thank you very much for the comment. Regarding the issue, we changed pressure injury care to pressure injury management.
[Overall]
pressure ulcer nursing ?
Response:
Thank you very much for the comment. Regarding the issue, we changed pressure injury nursing to pressure injury management
[p.12; Discussion]
it should be formulated differently, e.g. implement prophylaxis and undertake therapeutic activities related to treatment
Response:
Thank you very much for the comment. Regarding the issue, we changed the sentence according to your comments.
=> However, the current pressure injury management education is mainly designed to implement prophylaxes and undertake therapeutic activities related to treatment in general hospitals.
[p.13; Conclusion]
Conclusions should be structured differently
Response:
Thank you very much for the comment. Regarding the issue, we changed the paragraph according to your comments.
=> This study was conducted to develop and provide a framework for improving pressure injury management competency in nurses working in nursing homes. The results of this study support the KTA model finding that the quality of care at such facilities depends on nurses’ pressure injury management competence. Nurses at such facilities need to have integrated thinking ability, an understanding of the environmental context, communication skills for efficient decision-making, and a vision for systematic improvement for patient pressure injuries to be successfully managed. This framework should serve as the primary data for professional guidelines for experienced and novice nurses at long-term care facilities. To enhance nurses’ competence, the educational program utilizing this framework should be extended and established as standardized pressure injury training. The results of this study are expected to improve the safety and quality of life of the elderly in nursing homes.
Round 2
Reviewer 3 Report
Thank you for following the instructions, however, in one suggestion we did not understand each other. As a reader and clinician, I would like to know What are the competences of Korean nurses in the prevention and treatment of pressure ulcers, such a sentence should be included in the discussion. Answer with a footnote [35] is not directed to the question for me, please clarify it in the text as I suggested earlier